# Contrast-agnostic Spinal Cord Segmentation: A Comparative Study of ConvNets and Vision Transformers

**Enamudram Naga Karthik**[1,2]                      NAGA-KARTHIK.ENAMUNDRAM@POLYMTL.CA

**Sandrine Bédard**[1]                               SANDRINE.BEDARD@POLYMTL.CA

**Jan Valosek**[1,2]                                 JAN.VALOSEK@POLYMTL.CA

**Sarath Chandar**[2,3]                              SARATH.CHANDAR@MILA.QUEBEC

**Julien Cohen-Adad**[1,2]                           JCOHEN@POLYMTL.CA

[1] *NeuroPoly, Institute of Biomedical Engineering, Polytechnique Montréal, Montréal, Québec, Canada*

[2] *Mila - Québec Artificial Intelligence Institute, Montréal, Québec, Canada*

[3] *Department of Computer Engineering, Polytechnique Montréal, Montréal, Québec, Canada*

## Abstract

The cross-sectional area (CSA) of the spinal cord (SC) computed from its segmentation is a relevant clinical biomarker for the diagnosis and monitoring of cord compression and atrophy. One key limitation of existing automatic methods is that their SC segmentations depend on the MRI contrast, resulting in different CSA across contrasts. Furthermore, these methods rely on CNNs, leaving a gap in the literature for exploring the performance of modern deep learning (DL) architectures. In this study, we extend our recent work (Bédard et al., 2023) by evaluating the contrast-agnostic SC segmentation capabilities of different classes of DL architectures, namely, ConvNeXt, vision transformers (ViTs), and hierarchical ViTs. We compared 7 different DL models using the open-source *Spine Generic* Database of healthy participants (n = 243) consisting of 6 MRI contrasts per participant. Given a fixed dataset size, our results show that CNNs produce robust SC segmentations across contrasts, followed by ConvNeXt, and hierarchical ViTs. This suggests that: (i) inductive biases such as learning hierarchical feature reprensentations via pooling (common in CNNs) are crucial for good performance on SC segmentation, and (ii) hierarchical ViTs that incorporate several CNN-based priors can perform similarly to pure CNN-based models[1].

**Keywords:** Spinal Cord, MRI, Contrasts, Segmentation, Deep Learning, CNNs, Vision Transformers

## 1. Introduction

The cross-sectional area (CSA) of the spinal cord (SC) is an important biomarker for assessing cord compression and atrophy in neurological diseases such as multiple sclerosis (MS). However, the existing methods for SC segmentation have a *key* limitation: the predicted segmentation depends on the type of input MRI contrast and its acquisition parameters, resulting in different SC CSA for different MRI contrasts (Gros et al., 2018). Furthermore, such methods are dominated by CNN-based approaches (Gros et al., 2018; Bédard et al., 2023), suggesting a gap in the literature to evaluate other deep learning (DL) architectures, namely, ConvNeXt and ViT-based approaches for SC segmentation.

There exist several studies comparing the performance of vision transformers (ViTs) and CNNs primarily focusing on classification tasks (Matsoukas et al., 2021; Deininger et al., 2022; Fanizzi et al., 2023). However, the conclusions from these studies are not generalizable

---

1. Code: https://github.com/sct-pipeline/contrast-agnostic-softseg-spinalcord/releases/tag/v.2.3.1

as they are heavily dependent on the: (i) type of task (classification/segmentation/detection), (ii) input modality (i.e. digital pathology/natural/medical images), (iii) dataset sizes, and (iv) initialization strategy (i.e. from scratch/pretrained). Therefore, in this work, given a small dataset of SC MRIs with multiple contrasts, we compared the performance of the modern DL architectures (namely, CNNs, ConvNeXT, and ViTs) for automatic SC segmentation and evaluated their ability to achieve contrast-agnostic SC segmentation.

## 2. Materials and Methods

**Dataset** We used the open-access Spine Generic Public Database[2] consisting of healthy participants scanned on 3T MRI scanners across 42 sites. It consists of 6 MRI contrasts (T1w, T2w, T2*w, MT-on, GRE-T1w, and DWI) with both isotropic ($\{0.8, 1\}$ mm$^3$) and anisotropic ($\{0.5, 0.9\} \times \{0.5, 0.9\} \times \{3, 5\}$ mm$^3$) resolutions for each participant. This dataset presents a diverse set of MRI contrasts per participant to evaluate the models' contrast-agnostic segmentation capabilities. The final dataset ($n = 243$) was split according to 60%/20%/20% train/validation/test splits, resulting in 145/49/49 participants with 870/294/294 3D volumes, respectively.

**Preprocessing** To eliminate the differences in CSA within the ground truth (GT) SC masks across contrasts, our preprocessing strategy produced a unique, *soft* GT mask averaged across all MRI contrasts (please see Section 2.2 of Bédard et al. (2023) for details).

**Training Protocol** We followed the SoftSeg (Gros et al., 2021) training strategy, treating the segmentation task as a regression problem, where, (i) we *do not* binarize the inputs fed to the model after data-augmentation and, (ii) instead of DiceLoss (Milletarì et al., 2016), we use adaptive wing loss (Wang et al., 2019) which penalizes higher errors at the SC boundary. For a given subject, each contrast is treated as an independent input during training as opposed to concatenating all 6 contrasts as channels in a multi-modal input. Extensive ablations comparing the effects of DiceLoss and adaptive wing loss towards the SC CSA can be found in Section 3.2 of Bédard et al. (2023).

## 3. Experiments and Results

**Models and Training Details** We compared 7 models spanning 3 different classes of DL architectures. For CNNs, we compared DeepSeg 2D (Gros et al., 2018), nnUNet (Isensee et al., 2021), and our recently-introduced contrast-agnostic model (Bédard et al., 2023). The DeepSeg 2D and contrast-agnostic models are both available in the open-source Spinal Cord Toolbox (De Leener et al., 2017) under the commands `sct_deepseg_sc` and `sct_deepseg -task seg_sc_contrast_agnostic` respectively. Then, we trained MedNeXt (Roy et al., 2023), a state-of-the-art ConvNeXt model designed for 3D medical images. Lastly, among ViTs, we compared UNETR (Hatamizadeh et al., 2021), SwinUNETR (Hatamizadeh et al., 2022), and an open-source, pretrained SwinUNETR[3]. Except for the pretrained model which we fine-tuned on our datasets, all other models were trained from scratch for 200 epochs using Adam optimizer with a learning rate of 0.001 and a batch size of 2.

---

2. https://github.com/spine-generic/data-multi-subject/releases/tag/r20231212

3. https://github.com/Project-MONAI/tutorials/self_supervised_pretraining/swinunetr_pretrained

**Evaluation**  For quantitative assessment of the variability of CSA across contrasts, we used the CSA averaged across individual slices from the C2-C3 vertebral levels as the primary metric. Specifically, we (i) obtained the model predictions for each contrast, (ii) computed the absolute error between the CSA of the model prediction and the GT for each contrast, and finally, (iii) averaged the CSA errors across 6 contrasts for each subject (shown as one scatter point in the violin plot). The lower the absolute CSA error (in mm$^2$), the better, as, in theory, the spinal cord CSA value should *not* vary substantially across MRI contrasts for a given participant.

**Results**  Figure 3 shows the absolute CSA error across contrasts for each test participants across all models. Among the CNN-based models, the `contrast-agnostic` model achieved the lowest CSA error, with MedNeXt following closely. Interestingly, among the transformer models, UNETR showed the highest CSA error among all the models, while the SwinUNETR models showed similar performance as the CNNs.

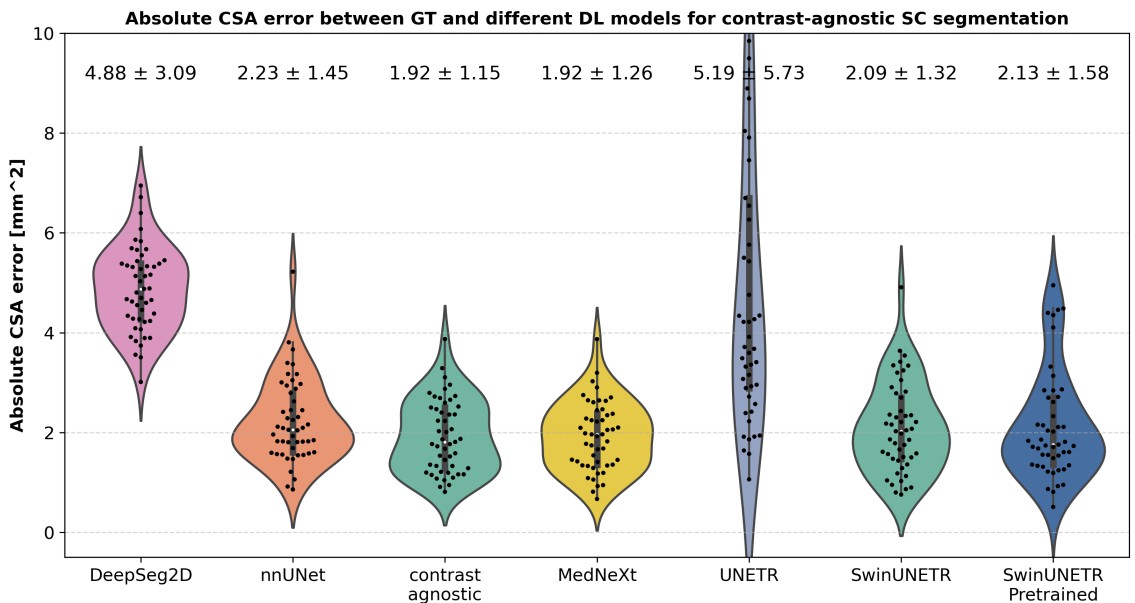

Figure 1: Absolute CSA error between the GT and predictions averaged across all 6 MRI contrasts for each model. Scatter plots within each violin show the CSA error averaged across all contrasts for a given participant. White triangle marker shows the mean CSA error across test participants.

## 4. Discussion and Conclusion

In this study, we performed a preliminary analysis of the performance of different classes of DL architectures for the specific task of contrast-agnostic SC segmentation. Overall, given a fixed dataset size, the CNN-based methods produce more robust SC segmentations across MRI contrasts. UNETR, which processes fixed-resolution 3D patches of size $16 \times 16 \times 16$ as 1D sequence of tokens performs the worst, suggesting that weak inductive biases in pure transformer-based encoders can be a major limiting factor for segmentation tasks. Hierarchical ViTs such as SwinUNETR that bring back CNN-based priors (e.g. learning

hierarchical representations via pooling and window-based local receptive fields, etc.) while using transformer blocks perform similar to CNNs. Future work aims at increasing the dataset size to include more contrasts and pathological images (such as MS) and comparing the performance of CNNs and SwinUNETR models at scale.

## Acknowledgments

This study was funded by funded by the Chair in Quantitative Magnetic Resonance Imaging [950-230815], CIHR [CIHR FDN-143263], the Canada Foundation for Innovation [32454, 34824], Fonds de Recherche du Québec (FRQ) - Santé [322736], NSERC of Canada [RGPIN-2019-07244], Canada First Research Excellence Fund, Courtois NeuroMod project, Quebec BioImaging Network [5886, 35450], INSPIRED (Spinal Research, UK; Wings for Life, Austria; Craig H. Neilsen Foundation, USA), Mila - Tech Transfer Funding Program, and National Institute of Neurological Disorders and Stroke of the National Institutes of Health (USA) [K23NS104211 and L30NS108301]. ENK is supported by FRQNT B2X Doctoral scholarship. SB is supported by the FRQNT Master's Training and the NSERC Graduate scholarships. JV received funding from the European Union's Horizon Europe research and innovation programme under the Marie Skłodowska-Curie Grant [101107932].

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
