# OpenReview forum: "Contrast-agnostic Spinal Cord Segmentation: A Comparative Study of ConvNets and Vision Transformers"
_MIDL.io/2024/Short_Papers — MIDL 2024 Short Papers_

### Official Review · Reviewer_3orw · 2024-04-15

**Confidence:** 4
**Final Rating:** 3.5

**Review:**

The work explores how CNNs and Transformers compare for segmentation of Spinal Cord and the calculation of its cross-sectional area. They find that convolutional networks work better than TransUnet (which does not have convolutional inductive biases), whereas architectures that re-introduce convolutional inductive biases to transformers (e.g. swin-unetr) perform similarly to convolutional nets.

Strenghts:
- The work performs a well executed benchmarking of the architectures.
- well written (mostly).
- There is a lot of discussion about new architectures claiming changes in SOTA. The paper takes a sober look and adds empirical evidence to the discussion about how well new architectures (transformer based) compare to the traditional convolutional nets, showing that the well established convnets are still doing well for vision, with modern transformer-based (with convolutional inductive biases builtin) doing similarly.

Weaknessses:
- The question has been previously explored, on different data, and the authors themselves claim that the findings do not necessarily translate between datasets/domains. And this paper benchmarks them on a single dataset, thus has the same limitation. But regardless, I think all together the evidence from different papers add to the pool of collective evidence that the community needs to collect over time.
- Some unclear points.


Other comments, which should be improved for camera ready in case paper is accepted:
- "However, the conclusions from these studies...": What were the conclusions from these studies? They are referenced, then stated that their conclusion do not generalise because they are on different data, but what are those "conclusions" has never been stated. I think it would be interesting to the reader, as this paper herein asks the same question (but on specific spine data).

- "contrast agnostic" segmentation => It is unclear in the paper how this is achieved. I am "guessing" that this is done by training a model using all 6 modality/contrast images at training, but only showing 1 random modality at each iteration for a subject (rather than all 6 as done in multi-modal models)? If this is true, it should be clarified somewhere (eg in the "training protocol" section).

- In "evaluation", the sentence "we computated the CSA averaged... for each test participant." is not clear. I don't understand if (a) each prediction is made using 1 modality and then an error for that prediction is calculated in comparison to the average of the 6 ground-truths. In this case, each violin plot shows 1 dot for each subject and for each modality? I.e. 6 dots per subject? Or.. (b) For each subject, 6 predictions are made, then the 6 predictions are averaged to get 1 CSA, and that average prediction is compared with the average of 6 ground truths? And in this case each violin plot has 1 error per subject? I think it is (b) but I am not sure. This is very important, and should be clarified. Please rephrase this sentence clearly and unambigously.

- "GT" undefined

Justiufication for recommendation:
Overall simple question, but ok executed benchmarking, and relatively well written paper. Could bring some discussion about sota architectures.

---

### Decision · Program_Chairs · 2024-04-26

Accept